# A Mikrokosmos of Mappings: An Empirical Study of YARRRML Evolution

Sehar Shafique*, Eduard Kamburjan

*IT University of Copenhagen, Copenhagen, Denmark*

**Abstract**

Knowledge graphs are increasingly constructed through declarative mapping languages, which allow users to define transformation rules from heterogeneous data sources into structured graph representations. As mappings evolve, assessing and maintaining their structural quality becomes a critical challenge. Yet no systematic framework for evaluating the quality of KG construction mappings exists. In this paper, we present the first empirical study of how YARRRML mappings evolve over time in open source development contexts, combining commit classification with a longitudinal analysis of six structural quality metrics adapted from software engineering. We analyze 521 commits across 35 open source YARRRML repositories and find (1) that feature additions and bug fixes dominate mapping evolution, (2) that repositories do not follow uniform growth and correction trajectories, mostly driven by changes in data and (3) URI and namespace errors are the most prevalent bug type. Our results indicate that support for dependencies in knowledge graph construction is lacking, and that declarative mappings must be seen as part of all data, software and knowledge ecosystems and lifecycles during development.

**Keywords**

Knowledge Graphs, YARRRML, Software Metrics, Repository Mining, Mapping Evolution

## 1. Introduction

Knowledge graphs (KG) are a prominent data representation model used in numerous fields such as biomedical research, transport infrastructure, and government open data initiatives. One widely used method to construct knowledge graphs is through *declarative mappings*, which define how semi-structured data from heterogeneous sources is transformed into a structured and interconnected representation. Languages such as RML, R2RML, and YARRRML are increasingly being adopted as the primary means of declarative construction of KGs [1, 2, 3], and as such one of the primary means users interact with knowledge graphs.

Declarative mappings evolve alongside the datasets and ontologies they depend on, yet their evolution and usage in practice remains underinvestigated. Their quality, however, cannot be ignored when assessing the created knowledge graphs: low quality mappings may produce incorrect or inconsistent knowledge graphs, directly affecting the downstream applications that depend on them [4]. Currently, mapping developers have no systematic way to assess whether their mappings are well-structured when maintaining them and no guidelines to follow when mappings grow large or become entangled. Similarly, tool developers have no overview which support is needed in practice.

Indeed, assessing the structural quality of declarative KG construction mappings remains an open challenge, due to their similarities to software: Declarative mappings define computational rules for transforming data and share structural characteristics with other software artifacts, such as dependencies, complexity, and modularity. Software engineering has developed a rich set of structural quality metrics for such artifacts, including complexity, cohesion, and coupling, yet no equivalent framework exists for KG construction mappings. This work investigates the evolution of KG construction mappings, grounded in an empirical study of how users structure, extend, and correct their mappings over time.

In this work, we focus on YARRRML [5], a human-readable syntax for RML [6] designed to lower the barrier to entry for mapping developers. Its emphasis on usability makes it particularly relevant for

*Second International Workshop on Users and Knowledge Graphs (UKG), co-located with SEMANTiCS'26: International Conference on Semantic Systems, September 15–17, 2026, Ghent, Belgium*

✉ sehars@itu.dk (S. Shafique); eduard.kamburjan@itu.dk (E. Kamburjan)

🆔 0009-0001-8101-0248 (S. Shafique); 0000-0002-0996-2543 (E. Kamburjan)

empirically investigating how mapping developers structure and evolve their mappings in open source contexts. We collect a corpus of open source YARRRML repositories and apply software engineering techniques to investigate how these mappings evolve over time. Specifically, we analyze the types of changes that developers make and the structural characteristics that evolve across the commit history of each repository.

To this end, we employ two techniques: commit classification based on the Conventional Commit Classification (CCC) framework [7] and a longitudinal analysis of structural quality metrics computed throughout the commit history of each repository. In doing so, we aim to shed light on whether YARRRML mappings evolve following patterns observed in software evolution, KG evolution, or represent different evolutionary trajectories. Concretely, we investigate the following research questions:

**RQ1** Do commit histories of open source YARRRML repositories contain sufficient information to detect and characterize mapping evolution patterns?

**RQ2** Which software quality metrics can be adapted to measure characteristics of YARRRML mappings, and do they produce meaningful variation across open-source repositories?

**RQ3** How do structural quality metrics of YARRRML mappings change across different types of developer commits?

**RQ4** What type of changes do developers make to YARRRML mappings in practice, and what evolution patterns emerge across open source repository histories?

Our empirical study of 35 open source YARRRML repositories and 403 classifiable commits reveal that mapping development is dominated by feature additions (44.4%) and bug fixes (37.7%), with URI and namespaces errors representing the most common identifiable bug type, which we interpret as a lack of dependency management support for declarative mappings. We identify three recurring evolution patterns across repositories and show that structural metrics produce meaningful variation, with refactor commits showing a clearer structural signal. Notably, semantic drift, changes driven entirely by external ontology and specification evolution, emerges as a novel category absent from conventional software commit taxonomies, and is structurally invisible to the metric set.

The contributions of this paper are threefold. (1) We present a curated dataset of open source YARRRML repositories with manually labeled commit histories, classified according to an extended version of the Conventional Commit Classification framework. (2) We present the first empirical investigation of the evolution and usability of declarative mappings in practice, characterizing the types of changes mapping developers make and the evolution patterns that emerge across repository histories. (3) We propose a set of structural quality metrics adapted from software engineering to measure the complexity, coupling, and size of YARRRML mappings.

The dataset is available under https://doi.org/10.5281/zenodo.20929556.

## 2. Background

### 2.1. Preliminaries

A YARRRML mapping file consists of one or more `mappings`, each of which is defined by a *triple map*. A triple map specifies a *logical source* (input data), a *subject* (the IRI template for the generated RDF resource), and a set of *predicate-object maps* (po), each of which defines a single output property of the mapping. Triple maps can reference one another through `parentTriplesMap` and `joinCondition` references, which establish dependencies between triple maps.

Figure 1 shows an example YARRRML mapping with two triple maps. The mapping `author` defines a *triple map* that reads from a *logical source* (`authors.csv`), generates a *subject* IRI using the `id` column, and produces three triples per row via three *predicate-object maps*: a type assertion, a name, and a date of birth. The mapping `publication` defines a second *triple map* that references `author` through a *join condition* on the `author_id` column, establishing a dependency between the two triple maps. The terms

```
  YARRRML
1  prefixes:
2  ...
3  sources:
4   authors-source:
5    access: authors.csv
6    referenceFormulation: csv
7   publications-source:
8    access: publications.csv
9    referenceFormulation: csv
10 mappings:
11  author:
12   sources: [authors-source]
13   s: ex:author/\$(id)
14   po:
15   ...
16   - [schema:birthDate,
17      \$(birth\_date), xsd:date]
```

```
  YARRRML
18
19  publication:
20   sources: [publications-source]
21   s: ex:publication/\$(pub\_id)
22   po:
23   - [a, schema:Book~iri]
24   - [schema:name, \$(title)]
25   - [schema:datePublished, \$(year),
26      xsd:date]
27   - p: schema:author
28     o:
29     - mapping: author
30       condition:
31       function: equal
32       parameters:
33         - [str1, \$(author\_id)]
34         - [str2, \$(id)]
```

**Figure 1:** An example YARRRML mapping illustrating its key structural component: prefixes, source, triple maps, and predicate-object maps.

*triple map*, *predicate-object map*, *logical source*, *subject map*, and *join condition* are used throughout this paper to refer to these structural components.

**Declarative Mappings** Declarative mapping languages provide a formal means of specifying how heterogeneous data sources are transformed into RDF-based knowledge graphs. Rather than encoding transformation logic imperatively, these languages allow developers to express mapping rules as structured declarations, separating the transformation specification from its execution. Several mapping languages have been proposed [8] each with different design goals and levels of expressivity. R2RML [9] is the W3C standard for mapping relational databases to RDF, defining the notion of a *triplesmap* as the core unit of a mapping, a rule that specifies a subject, a set of predicate-object pairs, and a logical source. RML [6] extends R2RML to support heterogeneous data formats including CSV, JSON and XML, while preserving the same structural model. YARRRML [5] is a human-readable serialization of RML expressed in YAML, designed be more accessible to practitioners. In this work, we focus on YARRRML as the target language, due to its emphasis on usability.

**Software Repository Mining** Software repository mining is an empirical research discipline that analyses the artifacts stored in version control systems, such as source code, commit histories, and issue trackers, to extract insights about software development practices and evolution. By examining how software changes over time, researchers can identify patterns in developer behavior, assess code quality and study the lifecycle of software projects [10].

## 2.2. Related Work

**Evolution.** Software evolution has been studied extensively in software engineering, with foundational results establishing that software systems must continuously change to remain useful, and that such change inevitably leads to increasing complexity and declining quality in the absence of deliberate maintenance effort [11, 12]. Declarative mappings have been established as an important and widely adopted component of knowledge graph ecosystems [13], used across a broad range of domains and deployment contexts. Mappings are sometimes generated automatically rather than hand-authored [14], which may have implications for their evolution patterns and the types of changes that appear in their commit histories, automatically generated mappings may exhibit different structural characteristics

and fewer manual corrections than those written and maintained by developers directly.

The evolution of knowledge graphs and ontologies has more recently attracted significant research attention within the semantic web community. Ontologies and knowledge graphs are not static artifacts, they are continuously refined as domains change, new knowledge emerges, and modeling decisions are revised [15]. Recent work has systematically characterized how knowledge evolves in open knowledge graphs, examining the types of changes that occur, their frequency and their propagation to dependent artifacts and proposing graph metrics to analyze such evolution. Knowledge graph evolution is defined as the process by which KGs change over time, encompassing the addition and removal of nodes and edges, while ontology evolution refers to the timely adaptation of an ontology and the consistent propagation of such changes to dependent artifacts [4].

The evolution of KG construction mappings has not been empirically studied. Unlike conventional software, mappings depend on external ontologies and data sources, which means they can change due to ontology updates, data sources modifications, or specification changes. These changes are driven by external factors rather than deliberate developer decisions. This paper presents the first empirical study of how YARRRML mappings evolve in open source projects.

**Empirical Studies.** Empirical software engineering applies scientific methods to the study of software development practices, using evidence gathered from real-world systems to validate or refute hypotheses about how software is engineered and maintained. Mining software repositories (MSR) is a subfield of empirical software engineering concerned with the extraction and analysis of artifacts stored in version control systems, including commit histories, issue trackers, and code changes, to identify patterns in developer behavior and software evolution [16]. A significant body of MSR research has focused on commit classification, that is, the problem of categorizing commits according to the type of change they represent. The Conventional Commit Classification (CCC) framework [17] provides a taxonomy of commit types including feature additions, bug fixes, refactoring, documentation, and maintenance tasks, validated across a large corpus of open source projects. Such classifications have been applied to study maintenance patterns, bug introduction rates, and the relationship between commit types and software quality indicators [18, 19].

Mining over ontology repositories or portals has been applied in the semantic web to collect ontologies to investigate, e.g., knowledge graph evolution [20], or different analysis of ontology pattern [21, 22, 23]. The connection to software has been investigated by Seifer et al. [24], who investigate SPARQL usage patterns in Java projects on github. To the best of our knowledge, no prior work has applied empirical methods to investigate evolution of declarative KG construction mappings, and their use in practice. Existing studies of mapping quality focus on the correctness and completeness of the output knowledge graph rather than the structural properties of the mapping artifact itself. This work addresses this gap by applying commit classification and structural metric analysis to a corpus of open source YARRRML repositories, providing the first empirical characterization of mapping development practices.

## 3. Methodology

This section describes the methodology used to conduct our empirical study. In Section 3.1, we describe the process used to construct a corpus of open source YARRRML repositories mined from GitHub using a semi-automated pipeline. In Section 3.2, we present the commit taxonomy used to classify mapping changes, extending the Conventional Commit Classification framework. Finally, in Section 3.3, we describe the six structural quality metrics adopted from object-oriented software engineering to measure the size, coupling, and dependency structure of YARRRML mappings.

### 3.1. Dataset Construction

This section discusses the process used to construct the dataset. We develop a corpus of open source YARRRML repositories from GitHub using an automated pipeline, illustrated in fig. 2. In Phase 1, GitHub

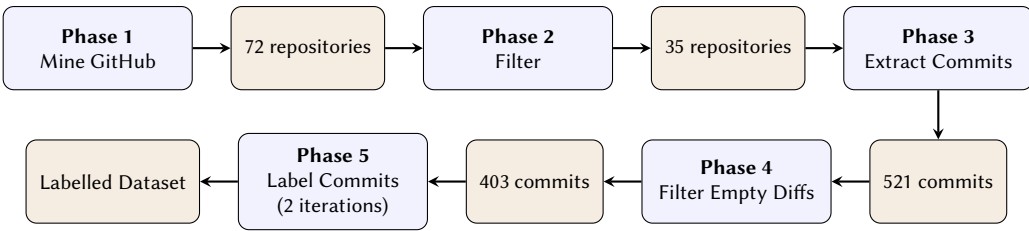

**Figure 2:** Dataset construction and commit labelling pipeline.

is searched using the API and the keyword `referenceFormulation` in YAML files, a term specific to YARRRML mappings, retrieving an initial set of 72 candidate repositories. In Phase 2, a keyword, filter is applied to remove tool and library repositories, and repositories without non-trivial commit histories are excluded, reducing the set to 35 repositories. In Phase 3, the 35 repositories are cloned locally and their commit histories are extracted, yielding 521 commits. In Phase 4, commits with empty mapping file diffs are removed, leaving 403 classifiable commits. In Phase 5, the remaining commits are manually labelled following the protocol described in section 3.3. The final corpus comprises 35 YARRRML repositories containing 521 commits across 717 diverse mapping files.

Inclusion criteria are defined to determine which repositories to include. A repository is included if it meets all the following criteria: it contains YARRRML mapping files, confirmed by the presence of the `referenceFormulation` keywords in YAML files, and has a non-trivial commit history, i.e., the YARRRML file is modified at least once after creation. Additionally, the repository must not be a fork or archived. Repositories are excluded if they are teaching repositories as they do not reflect real-world mapping practices and provide limited insight into how mappings evolve in practice.

## 3.2. Commit Taxonomy

We use the Conventional Commits Classification (CCC) [7] to label our commits. Conventional commits is a lightweight protocol for classifying commit messages into human and machine interpretable format. It defines a set of commit types such as fix, feat, refactor, docs, chore, and others, that express the reason behind each change, which are used in commit messages to indicate the intended type of the commit. Conventional Commits Classification is seeing increasing adoption in open source projects [17].

We extend CCC for repositories declarative mappings with one additional category known as semantic drift [4], capturing the commits where mappings have evolved because the ontology they have referenced has changed. This category is unique to the mapping domain and has no equivalent in conventional software commits. The full taxonomy is presented in Table 1.

## 3.3. Commit Labelling

Commit labelling was conducted in two iterations. In the first iteration, 100 commits were labelled, of which 74 contained non-empty mapping file diffs. The initial inter-annotator agreement, measured using Cohen's Kappa [25], was 0.1268 with a 95% confidence interval of [-0.010, 0.263], indicating slight agreement [25]. Following a discussion of disagreement and a refinement of the labelling guideline, a second iteration was conducted on 180 commits, of which 126 were non-empty. The agreed protocol proceeds as follows: the commit message is read first and assigned to a category if it clearly expresses the intent of the change. If the commit message is ambiguous or vague, the patch is inspected directly by examining the added and removed lines in the mapping file diff, specifically looking at whether new triple maps or predicate-object maps were added, whether existing ones were modified or removed, whether namespaces prefixes or URIs changed, and whether the change was triggered by an external ontology or data source update. Based on this inspection, the most appropriate category is assigned. A commit is assigned to the semantic drift category if the change is driven by an external ontology or specification update rather than deliberate developer intent. A commit is assigned to the unknown category if the mapping file diff is empty. In cases of ambiguity between two categories, the category

| Type | Description | CCC |
|---:|---|:---:|
| Fix | Corrects an error or bug in the mapping | ✓ |
| Feat | Adds new triple rules or extends existing mapping rules | ✓ |
| Refactor | Restructures mapping without changing RDF output | ✓ |
| Docs | Updates comments or documentation related to the mapping | ✓ |
| Chore | Version bumps, dependency updates, configuration changes | ✓ |
| Drift | Mapping changes due to ontology or data source evolution | Extension |
| Unknown | Could not be mapped to any defined category | ✓ |

**Table 1**
Commit Classification Taxonomy based on Conventional Commits

that best reflects the primary intent of the commit is chosen. The kappa score improved to 0.7069 with a 95% confidence interval of [0.602, 0.8117], indicating substantial agreement. The remaining commits were labelled by the first author following the refined protocol.

### 3.4. Chosen Metrics

Software quality metrics are measurable characteristics of software artifacts that capture structural features such as complexity, size, and coupling. They systematically assess the quality of software, support developers in determining maintainability issues, detect design problems, and make informed engineering decisions [26]. For instance, cyclomatic complexity measures the number of independent paths through a program, providing an indicator of how difficult a module is to test and maintain. However, the design and application of software metrics requires care, metrics are context-dependent and can be misinterpreted, or fail to capture the phenomena they are intended to measure [27, 28].

For KG construction mappings, we adapt six software quality metrics originally defined for classes in object-oriented programming [27], limiting this transfer to metrics that capture size and dependency structure, the two properties that triple maps and object-oriented (OO) classes share. All related predicate object mappings are grouped by triple map and have a defined identity, similar to a class in object-oriented programming. A triple map can also depend on other triple maps through `rr:parentTriplesMap` references and `rr:joinCondition`, reflecting the dependency structure between classes [29]. This correspondence justifies the transfer of structural metrics from object-oriented programming to YARRRML. However, we limit this transfer to metrics that capture size and dependency structure, the two properties that triple maps and OO classes share. Metrics that rely on properties specific to object-oriented programming, such as inheritance hierarchies or method-level characteristics, are not considered, as these have no meaningful counterpart in YARRRML mappings.

**Triple Map Count (TMC).** This metric measures the size of a mapping file in terms of declared `rr:TriplesMap` instances and corresponds to the number of classes declared within a software module. A mapping file with a larger number of triple maps is likely to be more complex and difficult to maintain and understand.

**PredicateObjectMap Count (POMC).** This metric measures the size of an individual triple map by counting the number of `rr:predicateObjectMap` declarations it contains. It is analogous to the number of methods in a class. It is computed per triple map across the full mapping file.

**Lines of Code (LoC).** This metric measures the size of a mapping file by counting the total number of lines in the YARRRML source file. Unlike TMC, which captures structural size in terms of declared triple maps, LoC captures the overall verbosity of the mapping regardless of its structural organization. It is computed directly on the raw YARRRML file before conversion to RML.

**Efferent Coupling (EC).** This metric counts the number of different triple maps that a given triple map depends on as defined by the dependency structure introduced above. A high efferent

| Commit Type | Count | Percentage |
|---|---|---|
| Feature (feat) | 179 | 44.4% |
| Bug Fix (fix) | 152 | 37.7% |
| Refactor | 34 | 8.4% |
| Chore | 22 | 5.5% |
| Semantic Drift | 14 | 3.5% |
| Documentation (docs) | 2 | 0.5% |
| **Total** | **403** | **100%** |

**Table 2**
Distribution of commit types across 403 classifiable commits.

coupling value indicates strong outward dependencies, increasing the triple map's sensitivity to structural changes elsewhere in the mapping.

**Afferent Coupling (AC).** This metric counts how many triple maps depend on a given triple map, reflecting its centrality within the mapping structure and the potential impact of modifying it.

**Dependency Depth (DD).** This metric measures the length of the longest dependency chain reachable from a given triple map, as defined above. For example, if triple map A depends on B which depends on C, the depth is 2. It is computed recursively until no further dependencies are found.

## 4. Data Analysis

**General Statistics.** Table 2 presents the distribution of commit types across the 403 classifiable commits. Feature additions and bug fixes together account for 81.9% of all classifiable commits, indicating that YARRRML mappings are primarily subject to growth and correction activity. Refactoring, maintenance, and documentation changes are comparatively rare, collectively accounting for 14.4% of commits. Semantic drift, changes driven entirely by external ontology or specification evolution, represent 3.5% of commits.

### 4.1. Commit Type Patterns

Table 2 shows the distribution of commit types across the 35 repositories. Feature additions dominate the corpus, accounting for 44.4% of all classifiable commits, followed by bug fixes at 37.7%. The remaining commit types, refactoring, chore, semantic drift, and documentation, collectively account for 17.9% of commits. Across repositories, two distinct evolutionary trajectories are observable. A subset of repositories exhibit a feat-dominant pattern, characterized by a feat/fix ratio greater than 1.0, indicating that mapping growth outpaces correction activity, Examples include `beltrans-data-integration` (ratio 1.76), `prelib-to-rdf(` ratio 1.88), and `phenopackets-v2-rdf-schema` (ratio 2.00). A second subset exhibits a fix-dominant pattern, with a feat/fix ratio below 1.0, suggesting that correction activity outpaces new development. Examples include `CARE-SM-Implementation` (ratio 0.55), `CDE-semantic-model-implementations`(ratio 0.71), and `demo-biomedit-workflow` (ratio 0.25).

To illustrate the types of changes captured by our taxonomy, we present two representative examples. Commit `53f0ca0` in `CARE-SM-Implementation` corrects an incorrect function prefix across multiple triple maps, replacing `idlab-fn:isNull` with `idlab:isNull` and `idlab-fn:str` with `idlab:str`. This is representative of the URI and namespace error category, the most common identified bug type.

Fix commit `53f0ca0` — CARE-SM-Implementation

```
- function: idlab-fn:isNull
-   parameters: [idlab-fn:str, $(protocol_id)]
+ function: idlab:isNull
+   parameters: [idlab:str, $(protocol_id)]
```

Commit `af8e778` in `beltrans-data-integration` illustrates semantic drift: the predicate `schema:publisher` is replaced with `rdagroup1elements:publishersName` from the RDA vocabulary, and a new prefix is added. The mapping structure is unchanged which means no triple maps are added or removed, but the semantic meaning of the generated triples changes. This change was driven by a semantic distinction in the ontology rather than a bug or new feature, and produces near-zero metric deltas despite being a meaningful change to the output knowledge graphs.

Semantic drift commit `af8e778` — beltrans-data-integration

```
+   rdagroup1elements: "http://rdvocab.info/elements/"
-   - p: schema:publisher
+   - p: rdagroup1elements:publishersName
  o: $(publisherName)
```

## 4.2. Evolution Patterns

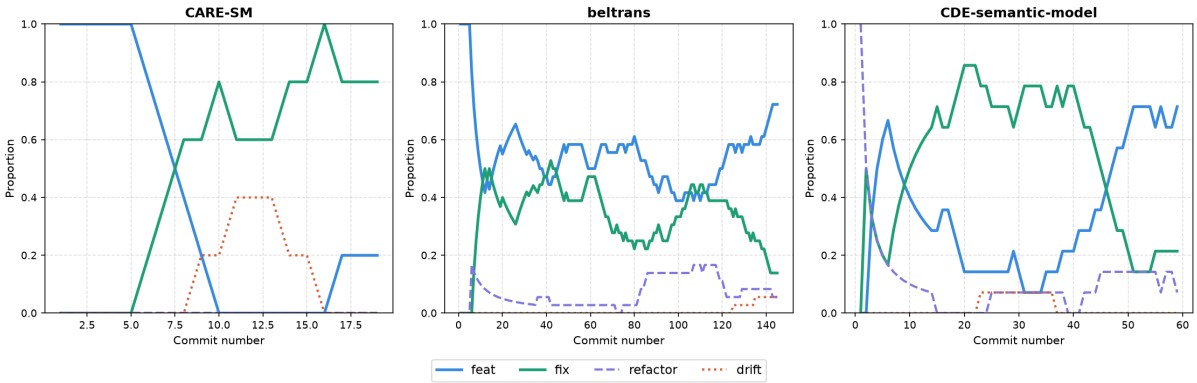

**Figure 3:** Rolling proportion of commit types across the commit histories of three representative repositories. The window size is proportional to the number of commits in each repository.

Figure 3 shows the rolling proportion of commit types across the commit histories of three representative repositories. Each repository follows a peculiar pattern. `CARE-SM-Implementation` starts with mostly feature additions, which then drop sharply as bug fixes take over, showing a clear build-then-fix trajectory. A cluster of semantic drift commits appears mid-history, driven by changes to the SIO Ontology specification. `beltrans-data-integration` shows a more stable pattern, with feature additions and bug fixes remaining roughly balanced throughout. `CDE-semantic-model-implementations` shows the opposite of CARE-SM, with bug fixes dominating early on before feature additions increase steadily in the later commits.

To further examine how commit types evolve within individual repositories, we divide each repository's commit history into two equal parts and compare the proportion of `feat` and `fix` commits across each half. The three repositories were selected to represent the particular evolutionary trajectories observed across the corpus. Table 3 presents the results for repositories with at least ten classifiable commits. The most pronounced temporal shift is observed in `CARE-SM-Implementation`, where the proportion of feat commits decreases from 55.6% in the first half to 10.0% in the second half, while

| Repository | feat% (1st) | feat% (2nd) | fix% (1st) | fix% (2nd) |
|---|---|---|---|---|
| beltrans | 54.2 | 57.5 | 36.1 | 27.4 |
| CDE-semantic-model | 24.1 | 50.0 | 65.5 | 40.0 |
| MapToMethod | 38.5 | 30.8 | 30.8 | 23.1 |
| prelib-to-rdf | 61.5 | 53.8 | 15.4 | 46.2 |
| CARE-SM | 55.6 | 10.0 | 33.3 | 80.0 |
| phenopackets-v2 | 57.1 | 57.1 | 14.3 | 42.9 |
| era-data-mappings | 50.0 | 33.3 | 50.0 | 50.0 |
| social-media | 40.0 | 50.0 | 40.0 | 0.0 |
| **Average** | 47.6 | 42.8 | 35.7 | 38.7 |
| **Median** | 52.1 | 51.9 | 35.5 | 41.5 |

**Table 3**

Temporal evolution of `feat` and `fix` commit proportions across repository histories

fix commits increase from 33.3% to 80.0%. This pattern is consistent with a build-then-fix trajectory in which early development is dominated by feature additions, followed by a correction phase as the mapping matures. On average, the proportion of feat commits decreases slightly from 47.6% in the first half to 42.8% in the second half, while the proportion of fix commits increases from 35.7% to 38.7%, suggesting a modest but consistent shift towards correction activity as repositories mature.

Bases on these observations, we identify three recurring evolution patterns. Pattern I is characterized by an initial growth phase dominated by feature additions followed by a correction phase dominated by feature additions followed by a correction phase dominated by bug fixes, as observed in `CARE-SM-Implementation`. Pattern II is characterized by a consistently balanced mix of feature additions and bug fixes throughout the repository history, as observed in `beltrans-data-integration`. Pattern III is characterized by an early correction phase followed by renewed feature development, as observed in `CDE-semantic-model-implementation`.

To complement the commit type analysis, Fig 4 presents the evolution of structural metrics over time for three repositories that have non-zero coupling metrics, as the majority of repositories in the corpus exhibit no inter-triple-map dependencies. `phenopackets-v2-rdf-schema` shows rapid growth in TMC during the first none commits before stabilizing, with DD remaining consistently high throughout, indicating that complex join dependencies were established early and maintained as the mapping grew. `prelib-to-rdf` maintains a relatively stable TMC, but AC and DD drop significantly around commits 12 to 15 before recovering, coinciding with refactoring commits where separate data sources were consolidated. `era-data-mappings` shows a dip in TMC around commits 4 to 5 before growing strongly, while DD spiked around commits 8 to 9 before dropping back to near zero, suggesting a period where complex join dependencies were temporarily introduced and restructured.

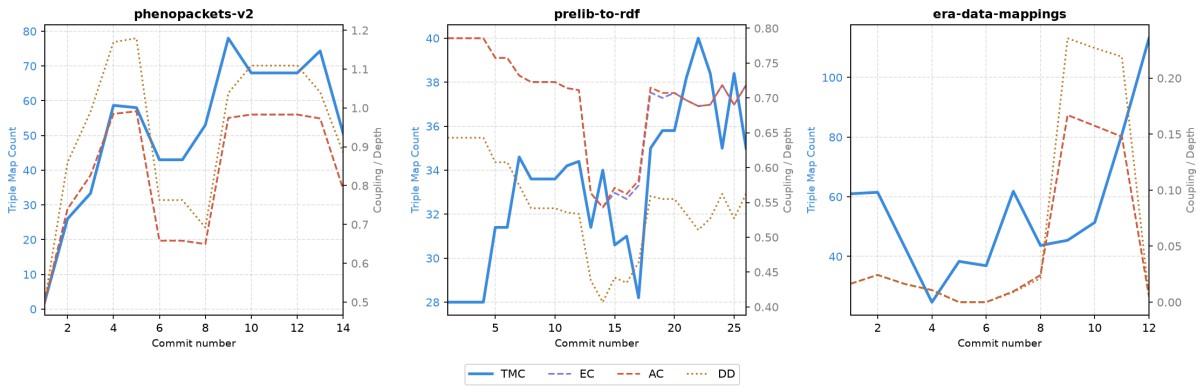

**Figure 4:** Evolution of TMC (left axis) and coupling metrics - EC, AC and DD (right axis) - across the commit histories of three repositories with non-zero coupling metrics

## 4.3. Other Commit Types

We focus on `feat` and `fix` commits as they collectively account for 81.9% of all classifiable commits and represent the primary drivers of mapping evolution, making them the most informative for detecting temporal patterns. Table 5 and 4 present the distribution of change types within `feat` and `fix` commits respectively. Among feat commits, new data source connections represent the most common change type (22.9%), followed by new predicates and triple maps (15.6%), new identifier and label mappings (11.7%) and new relationship joins (10.1%). Among fix commits, URI and namespace errors represent the most common identifiable bug type (17.1%), followed by wrong predicate or mapping rule errors individually accounting for 10.5% of fix commits.

**Semantic Drift**   Of the 403 classifiable commits, 14 are classified as semantic drift. These commits represent changes driven entirely by external ontology or specification evolution, rather than deliberate developer intent. Observed instances include namespace URI changes, ontology term replacements, and migrations to new versions of the RML specification. Table ?? presents the distribution of semantic drift commits by sub-type. The other category includes commits that are externally driven but do not fit nearly into the defined sub-types, such as changes made to align with updated modelling conventions in a referenced ontology, or revisions triggered by community feedback at domain-specific events.

| Bug Type | Count | Perc. |
|---|---|---|
| URI/IRI/namespace error | 26 | 17.1% |
| Wrong predicate/mapping rule | 16 | 10.5% |
| Wrong data source/path | 16 | 10.5% |
| Syntax/formatting error | 11 | 7.2% |
| Wrong type/datatype | 8 | 5.3% |
| Broken reference/link | 4 | 2.6% |
| Wrong column/field reference | 1 | 0.7% |
| Other | 70 | 46.1% |
| **Total** | **152** | **100%** |

**Table 4**

Types of bugs in fix commits

| Change Type | Count | Perc. |
|---|---|---|
| New data source connection | 41 | 22.9% |
| New predicate/triple map | 28 | 15.6% |
| New identifier/label mapping | 21 | 11.7% |
| New relationship/join | 18 | 10.1% |
| General new feature | 31 | 17.3% |
| New namespace/ontology term | 4 | 2.2% |
| Other | 36 | 20.1% |
| **Total** | **179** | **100%** |

**Table 5**

Types of changes in feat commits

**Metrics.**   Table 6 presents the mean metric delta per commit type, computed as the difference in metric values between consecutive commits within the same repository. Metric deltas are aggregated at the commit level by averaging across all triple maps in a given commit.

**Table 6**

Mean metric delta per commit type

| Type | TMC | POMC | EC | AC | DD |
|---|---|---|---|---|---|
| feat | +1.06 | -0.01 | +0.01 | +0.01 | +0.01 |
| fix | +0.83 | -0.02 | +0.00 | +0.00 | +0.01 |
| refactor | -6.35 | +0.88 | -0.05 | -0.05 | -0.04 |
| chore | +1.99 | -0.57 | -0.01 | -0.01 | -0.02 |
| drift | -0.85 | -0.17 | +0.00 | +0.00 | +0.00 |
| docs | +30.0 | -1.38 | +0.05 | +0.05 | +0.05 |

Feature commits show a small positive delta on TMC (+1.06), indicating that mappings grow slightly in size when new functionality is added. Bug fix commits show near-zero deltas across all metrics, suggesting that correcting errors does not alter the structural composition of a mapping. Refactor

commits show the most pronounced pattern, a large negative delta on TMC (-6.35) and negative delta across all coupling metrics, consistent with the expectation that refactoring reduces structural complexity. Semantic drift commits show near-zero deltas across all metrics, indicating that externally driven ontology and specification changes do not produce observable structural change.

# 5. Results and Discussion

This section presents the results of our empirical study and discusses their implications. We structure the results around the four research questions introduced above.

**RQ1: Do commit histories of open source YARRRML repositories contain sufficient mapping evolution patterns?**   Commit histories provide sufficient information to detect evolution patterns at the level of commit type distribution and temporal trajectories. We can extract commit messages, patches, and file-level diffs to classify changes, measure the proportion of `feat` and `fix` commits over time, and identify structural changes between consecutive commits. This enables the detection of evolution patterns I, II and III, as well as the presence of externally driven semantic drift.

However, 22.6% of commits produced no observable change in mapping files, and a significant proportion of commit messages are too vague to classify without inspecting the patch directly. Furthermore, a substantial fraction of commits produce no observable diff in the mapping files themselves, suggesting that not all mapping-related changes are captured at the file level. This indicates that commit messages in YARRRML repositories are often insufficiently descriptive to enable reliable classification without direct patch inspection. Only 10 out of 403 commits (2.5%) in our corpus follow the Conventional Commits Specification format, and only 3 out 35 repositories adopt it in any form. A study on the adoption of CCC [17] found that while a growing number of projects on GitHub are adopting the Conventional Commits Specification, with ~10% adoption for github software projects in 2023. CCC-type confusion remains the most common challenge developers face. In our YARRRML corpus, adoption is even more limited, only 2.5% of commits follow the CCC format, suggesting that the mapping community has not yet engaged with structured commit message practices at all, making both manual and automated classification more challenging than in mainstream software repositories.

**RQ2: Which software quality metrics can be adapted to measure structural characteristics of YARRRML mappings, and do they produce meaningful variation across open source repositories?**   We show that six structural metrics adapted from object-oriented programming (TMC, POMC, LoC, EC, AC and DD) can be successfully applied to YARRRML mappings. These metrics are applicable because triple maps share two fundamental properties with OO classes: they have a defined size in terms of declared elements, and they can depend on one another through `rr:parentTriplesMap` references, establishing a dependency structure analogous to class coupling. Across the 35 repositories in our corpus, the metrics produce meaningful variation. TMC ranges from 1 to over 100 triple maps per mapping file, reflecting the wide range of mapping complexity observed in practice. Coupling metrics similarly vary across repositories, with some mappings exhibiting no inter-triple-map dependencies and other showing deep dependency chains. The metrics reveal further insights, temporal analysis shows that refactor commits consistently reduce structural complexity while feature commits drive gradual growth, and semantic drift commits produce near-zero metric deltas despite representing meaningful semantic changes. These results confirm that the adoption of software quality metrics for YARRRML mappings is meaningful and produces actionable information about mapping structure and evolution.

**RQ3: How do structural quality metrics of YARRRML mappings change across different types of developer commits?**   As shown in table 6, different commit types produce distinct delta patterns. Refactor commits show the most pronounced structural signal, with a large negative delta on TMC (-6.35) and negative deltas across all coupling metrics, indicating that refactoring in YARRRML mappings reduces structural complexity. Feature commits show a small positive delta on TMC (+1.06), reflecting

gradual mapping growth. Bug fix commits show near-zero deltas across all metrics, indicating that correcting errors does not alter the structural complexity of a mapping. Notably, semantic drift change, revealing that ontology and specification driven changes are structurally invisible to the metrics alone.

**RQ4: What types of changes do developers make to YARRRML mappings in practice, and what evolution patterns emerge across open source repository histories?** Out of 403 classifiable commits, feature additions (44.4%) and bug fixes (37.7%) dominate mapping development, collectively accounting for 81.9% of all changes. Refactoring is comparatively rare (8.4%), and semantic drift, changes driven entirely by external ontology or specification evolution, accounts for (3.5%) of commits. Among feature additions, new data source connections are the most common change type (22.9%), indicating that mapping growth is primarily data-driven. Among bug fixes, URI and namespace errors are the most prevalent bug type (17.1%), reflecting the difficulties that arise from maintaining correct references to external ontologies as they evolve and their namespaces change over time.

The evolution analysis reveals that repositories follow diverse patterns of evolution. We identify three recurring patterns. *Pattern I* is characterized by an initial phase dominated by feature additions followed by a stabilization phase dominated by bug fixes, as observed in `CARE-SM-Implementation`. *Pattern II* is characterized by a consistently balanced mix of feature additions and bug fixes throughout the repository history, as observed in `beltrans-data-integration`. *Pattern III* is characterized by an early correction phase followed by renewed feature development, as observed in `CDE-semantic-model-implementations`. Together these patterns suggest that YARRRML mappings do not follow a single universal evolution. In particular, the presence of semantic drift commits aligns with the types of changes identified in studies of open knowledge graph evolution, where ontology updates and schema modifications propagate to dependent artifacts [4]

## 5.1. Insights

Our empirical study yields the following insights about YARRRML mapping development. First, mapping development shares characteristics with software development but also exhibits domain-specific patterns. The dominance of `feat` and `fix` commits mirrors patterns observed in software repositories, and evolution pattern observed across repositories are consistent with foundational results in software evolution [12]. However, the prevalence of URI and namespace errors as primary bug type, reflecting the lack of dependency management in YARRRML projects, and the relatively small proportion of semantic drift commits (3.5%), suggest that mapping evolution is not primarily driven by external ontology chanhes but follows patterns more characteristic of software development.

> **Insight 1: URI and Namespace Errors are the Primary Bug Type**
>
> The most common bug in YARRRML mappings is incorrect URI and namespace references. This points to the need for tooling support for dependency management specifically, automated validation of ontology references at commit time.

Second, structural metrics are insufficient to fully characterize mapping evolution. Semantic drift commits, which represent meaningful changes to the semantic output of a mapping, are structurally invisible, producing near-zero metric deltas. This finding suggests that a complete quality assessment framework for YARRRML mappings must combine structural metrics with commit classification to capture the full range of mapping changes.

> **Insight 2: Structural Metrics are Insufficient alone**
>
> Semantic drift commits are structurally invisble to the metric set, producing near-zero deltas despite representing meaningful semantic changes. A complete quality assessment framework for KG construction mappings must combine structural commit classification.

Third, mapping growth is primarily data-driven at the corpus level, with new data source connections representing the most common feat change type (22.9%). However, this pattern is not uniform across repositories, for instance in `beltrans-data-integration` new data sources account for 34.6% of feat commits, while in `CARE-SM-Implementation` and `CDE-semantic-model-implementations` general feature additions and other changes dominate, suggesting that growth patterns vary depending on the nature and purpose of the mapping project.

> **Insight 3: Declarative Mappings are Connected to Data, Software and Knowledge**
>
> YARRRML mappings are connected to the lifecycles and ecosystems for data, software and knowledge: The most prevalent bug is knowledge-based (wrong URIs), evolution is driven in a big part by changes in input data, yet the evolution patterns themselves mirror evolution of software.

### 5.2. Threats to Validity

**Construction Validity.** The structural metrics used in this study are adopted from object-oriented software engineering and may not fully capture the semantic properties of YARRRML mappings. In particular, metrics based on `rr:parentTiplesMap` references may underestimate coupling in mappings that use alternative join mechanisms. Additionally, the conversion of YARRRML files to RML via yarrrml parser introduces a processing step that may fail for malformed or non-standard YARRRML syntax, resulting in 29 files being skipped during metric computation. The commit classification relies on manual labelling, which introduces subjectivity despite the inter-annotator agreement of 0.7069 achieved after two iterations.

**Internal Validity.** The temporal analysis divides each repository's commit history into two equal halves, which is a simplification that may not capture more complex evolution patterns. The keyword-based categorization of commit messages into sub-types such as URI errors or new data source connections is based on surface-level matching and may misclassify commits with ambiguous or vague messages. The 22.6% of commits excluded as unknown due to empty mapping file diffs may introduce bias if these commits are systematically different from classifiable commits.

**External Validity.** The corpus comprises 35 open source YARRRML repositories, which may not be representative of all YARRRML mapping projects, particularly closed source or industrial deployments. The findings are specific to YARRRML and may not generalize to other declarative mapping languages such as R2RML or RML in Turtle syntax. Future work should extend the study to a broader corpus of mapping repositories across multiple languages to validate the applicability of the identified evolution patterns and metric behaviors.

## 6. Conclusion

KG construction mappings are increasingly central to knowledge graph ecosystems, yet their structural quality and evolution remain poorly understood. In this paper, we present the first empirical study of how YARRRML mappings evolve in open source development contexts. We contributed a curated dataset of 35 repositories with 403 labelled commits, a set of six structural quality metrics adapted from object-oriented software engineering, and an extended commit taxonomy that introduces semantic drift as a novel category specific to the mapping domain.

Our results show that mapping development is dominated by feature additions (44.4%) and bug fixes (37.7%), with URI and namespace errors representing the most common identifiable bug type, reflecting the challenges of maintaining correct dependencies on external ontologies. We identified three recurring evolution patterns across repositories, Pattern I (build-then-fix), Pattern II (sustained development), and Pattern III (fix-then-growth), suggesting that YARRRML mappings exhibit characteristics of both

software and knowledge graph evolution rather than following a single universal model. Structural metrics produce meaningful variation across repositories, with refactor commits showing the clearest structural signal through reductions in Triple Map Count and coupling metrics, while semantic drift remain structurally invisible to the metric set.

Our findings suggest that when studying the evolution KG construction mappings, all three dimensions must be considered: software evolution patterns, knowledge graph evolution patterns, and ontology evolution patterns. Mappings sit at the intersection of all three, they grow and are corrected like software, they are coupled to ontologies that change independently, and they reflect the evolving state of the knowledge graph they produce. A complete quality framework of KG construction mappings must account for all three dimensions.

This study is limited to YARRRML and open source repositories, and future work should extend the analysis to other declarative mapping languages such as R2RML and RML and closed source deployments to validate the generalization of the identified phenomena. We also aim to investigate the relationship between mapping quality and the quality of the knowledge graphs they produce.

**Acknowledgments**    This work is supported by the DFF project *Graph-based Verification of Reflective Programs* (5254-00016B).

## Declaration on Generative AI

During the preparation of this work, the authors used Claude (Anthropic) in order to: check grammar, improve language, and improve readability. After using this tool, the authors reviewed and edited the content as needed and takes full responsibility for the publication's content.

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
