# OpenReview forum: "A Mikrokosmos of Mappings: An Empirical Study of YARRRML Evolution"
_SEMANTiCS.cc/2026/Workshop/UKG — SEMANTiCS 2026 Workshop UKG Submission_

### Official Review · ~Jakub_Duchateau1 · 2026-07-13
**Valuable insights but limited confidence due to dataset and reproducibility issues**

**Rating:** 7
**Confidence:** 4

**Review:**

This paper presents an original and relevant empirical study on YARRRML mapping evolution across 35 repositories, proposing six metrics transposed from object-oriented software engineering and a novel "semantic drift" commit class in Conventional Commits. These insights are valuable for ecosystem tooling, but I have some concerns regarding dataset reproducibility, and the reliability of the manual commit classifications. Provided these methodological and reproducibility issues are addressed, this will be a strong contribution.

- The provided scope justification, YARRRML "emphasis on usability", feels anecdotal and disconnected from the paper's actual objectives. Furthermore, the usability of YARRRML measured in https://doi.org/10.7717/peerj-cs.318 suggests caution with this claim.
- The repository collection methodology in its second phase mentions filtering out "tool and library repositories". However, looking at the dataset, it appears to have missed several tool repositories: `citiususc/yatter`, `RMLio/yarrrml-parser`, `oeg-upm/yarrrml-validation`, `morph-kgc/morph-kgc`.
- **Unclear misclassification**: The authors highlight the commit `53f0ca0` changing `idlab-fn` to `idlab` as their prime example of a bug fix, when this is arguably a namespace rename; therefore, it seems to correspond to the definition of a refactoring commit. This raises concerns about how strictly we can trust the manual classifications.
- The provided dataset script relies on undocumented dependencies: the `gh` CLI and `git`, which download the data from GitHub; however, rewriting history, making repos private or version changes in `gh` could make the script non-functional. Providing an archive of the raw commits on Zenodo would ensure long-term reproducibility.
- The metrics are detailed enough that reimplementing them should be possible, but the queries/programs computing them should be published to facilitate reproducibility.
- **Overstated Conclusions**: The authors claim, "Bases on these observations, we identify three recurring evolution patterns." (p. 9) then in conclusions "We identified three recurring evolution patterns across repositories" (p. 13), but this conclusion appears to be heavily derived from the analysis of only three specific repositories. Calling these "recurring patterns" across the ecosystem based on such a small sample is slightly overconfident, but maybe I have this impression due to the formulation.

Then I have some suggestions, minor comments, and typos:

- Regarding the insight that developers lack tooling for dependency management and ontology validation at commit time, it could be mentioned that existing efforts in this space exist, such as GRAPE or the Semantic Web Language Server (SWLS) that have varying levels of support for it.
- Mention the tool used for analysing/labelling the commits because working with that big JSONL must have been very impractical.
- Figure 1 contains LaTeX glitches (e.g., `\$(birth\_date)`).
- Using arXiv papers [27, 28] to support established conclusions about software metrics is not ideal, especially those as finding a reviewed article should be possible.
- **Missing table "Distribution of semantic drift commits by sub-type" in Semantic Drift paragraph**. (Confirmed by email that it was a LaTeX error.)
- "CARE-SM-Implementation" vs "CARE-SM".
- Typo: "chanhes" → changes.
- RQ1 could be more clear. Is it about patterns or is it about the possibility of establishing patterns from the commits data ?
- "The mapping structure is unchanged which means no triple maps are added or removed" Is structure only limited to TM addition/removal?
- `other_modified_files` is always empty on my side when checking the data.
- **Missing subclassification data in the dataset**. (Confirmed by email, the data exists and will be added.)
- "A complete quality assessment framework for KG construction mappings must combine structural commit classification." (Insight 2) is the sentence complete? combine with...

---

### Official Review · ~Mario_Scrocca1 · 2026-07-21
**Good paper on underexplored topic, fit to workshop could be improved**

**Rating:** 7
**Confidence:** 4

**Review:**

The paper addresses an underexplored and relevant topic by analysing the evolution of declarative mappings for KG construction, with a focus on YARRRML. The empirical analysis of open-source repositories, commit types, structural metrics, and emerging evolution patterns provides useful evidence on how mappings evolve in practice. Moreover, the dataset published by the authors represents a valuable resource that could be further analysed by other researchers and practitioners. I also appreciated the "threats to validity" analysis in Section 5.2, which clearly acknowledges several limitations of the work.

The fit with the workshop is good, although it could have been strengthened by discussing more explicitly what the results imply for users. For instance, the prevalence of URI and namespace errors suggests the need for validation tools during mapping authoring that can signal inconsistencies with target ontologies. It would also have been interesting to contact users or maintainers of the analysed repositories to validate whether the inferred commit classifications and evolution patterns correspond to their actual development history, and whether specific contextual factors affected such history.

As a more specific comment, I have some concerns about the notion of semantic drift. The paper presents it as a mapping-specific category that does not exist in conventional software, but this distinction should be better justified. In software, changes caused by external services/APIs/libraries are also common and are often classified as fixes, chores, or features depending on their impact. I also miss a clearer explanation of how the authors distinguished between changes caused by the actual evolution of a target ontology and those due to earlier modelling errors discovered at a later stage (was the evolution of the referenced ontologies explicitly checked?). Related to that, changes to the RML mapping ontology should probably not be considered as semantic drift.

Overall, I believe this is a promising contribution that, despite being preliminary, highlights several relevant insights that could foster discussion during the workshop and open up directions for further investigation.

Minor comments:
- The paper should be proofread carefully. For (i) example: the sentence "due to their similarities to software: Declarative mappings…" in the Introduction should be rephrased; (ii) the description of Pattern I in Section 4.2 contains repeated text.
- Unresolved "Table ??" reference in Section 4.3 (and missing table?)
- In Section 2.1, the explanation of declarative mappings should come before the detailed explanation of YARRRML